

# Proteomics research and related functional classification of liquid sclerotial exudates of *Sclerotinia ginseng*

Dan Wang, Jun Fan Fu, Ru Jun Zhou, Zi Bo Li and Yu Jiao Xie

Department of Plant Protection, Shenyang Agricultural University, Shenyang, Liaoning, China

## ABSTRACT

*Sclerotinia ginseng* is a necrotrophic soil pathogen that mainly infects the root and basal stem of ginseng, causing serious commercial losses. Sclerotia, which are important in the fungal life cycle, are hard, asexual, resting structures that can survive in soil for several years. Generally, sclerotium development is accompanied by the exudation of droplets. Here, the yellowish droplets of *S. ginseng* were first examined by sodium dodecyl sulfate-polyacrylamide gel electrophoresis, and the proteome was identified by a combination of different analytical platforms. A total of 59 proteins were identified and classified into six categories: carbohydrate metabolism (39%), oxidation-reduction process (12%), transport and catabolism (5%), amino acid metabolism (3%), other functions (18%), and unknown protein (23%), which exhibited considerable differences in protein composition compared with droplets of *S. sclerotium*. In the carbohydrate metabolism group, several proteins were associated with sclerotium development, particularly fungal cell wall formation. The pathogenicity and virulence of the identified proteins are also discussed in this report. The findings of this study may improve our understanding of the function of exudate droplets as well as the life cycle and pathogenesis of *S. ginseng*.

## INTRODUCTION

Ginseng (*Panax ginseng* C. A. Meyer) is a perennial medicinal herb in the Araliaceae family that is mainly distributed in China, North Korea, South Korea, Japan and Russia. As a precious medicinal material, it is widely used in the fields of medical care, health care, and the chemical industry. It is highly valued because its medicinal ingredients have extracts containing ginsenosides (*Sun, 2011*; *Wan et al., 2015*; *Wang et al., 2016*), essential oil, polysaccharides (*Wan et al., 2015*), and peptides (*Sun, 2011*). These make ginseng possess antitumor (*Sun, 2011*; *Zhou et al., 2014*), immunoregulatory, antioxidant (*Sun, 2011*; *Yu et al., 2014*), antihyperglycemic (*Jiao et al., 2014*), and antimalarial activities (*Han et al., 2011*). China is the largest ginseng production area, with a yield of up to 70% of the world's output. In recent years, with the increasing recognition and demand of ginseng, the artificial cultivation area of ginseng has continuously expanded, which could easily lead to disease epidemic.

Because the roots are the main medicinal parts of ginseng, root disease by various soil pathogens including *Cylindrocarpon*, *Pythium*, and *Sclerotinia* (*Cho et al., 2013*) cause

Corresponding author
Jun Fan Fu, fujunfan@163.com

inestimable economic losses. In fact, *S. ginseng* has already given rise to a serious epidemic disease in northeast China, with an incidence ranging from 10% to 15%, and up to 20% in severe cases. Mycelium is the main source of infection and can directly infect the root and basal part of the ginseng stem. Symptoms of *S. ginseng* infection first includes a rusty brown-colored epidermis, water-soaked lesions, and dissolved pith, which eventually leave the ginseng epidermis; later, sclerotia can be found on the surface of the infected root tissues and basal part of the stems. Sclerotia are hard, asexual, resting structures (*Erental, Dickman & Yarden, 2008*) that use overwintering as a way to survive in soil for several years (*Adams & Ayers, 1979*; *Kwon et al., 2014*). Under adverse conditions, the mycelium of *S. ginseng* stop its vegetative growth and begin to coalesce into sclerotia. Sclerotia, as common dissemination structures of many important agricultural crop pathogens, such as *S. sclerotiorum*, *Sclerotium rolfsii*, *B. cinerea*, and *Claviceps purpurea* (*Erental, Dickman & Yarden, 2008*), play important roles in their niche.

Sclerotia development is usually divided into three stages: sclerotia initial (SI), sclerotia developing (SD), and sclerotia mature (SM) (*Patsoukis & Georgiou, 2007*). The formation of liquid droplets by sclerotia is a common feature during sclerotia development (*Cooke, 1969*; *Colotelo, 1973*; *Liang, Strelkov & Kav, 2010*). The very beginning of liquid droplet occurrence is observed on the surface of aerial hyphae at the initial stage, and the droplets increase in size during growth of the sclerotia. At the SD stage of development, exudate droplets can be clearly observed by the naked eye on the surface of interwoven hyphae. Concomitantly with mature features, including surface delimitation, internal consolidation, and pigmentation (*Erental, Dickman & Yarden, 2008*), the droplets reach maximum quantity but disappear upon further culturing. The dehydration and thickening of cell walls, polymerization of soluble compounds, and decreased moisture in tissues may all be reasons to condense the active exudation of water on the surface of sclerotia (*Daly, Knoche & Wiese, 1967*; *Willetts, 1971*; *Chet & Henis, 1975*; *Willetts & Bullock, 1992*). However, these droplets are only observed in culture and not under field conditions, possibly because of the combined effects of air-drying, the absorption of soil, and the recycling of exudates needed for sclerotia development (*Pandey et al., 2007*). After the droplets disappear, a copious dried-up deposit consisting of membranous material is left on the sclerotial surface (*Colotelo, Sumner & Voegelin, 1971*).

Exudation has a very complex composition, as it contains many kinds of substances such as soluble carbohydrates, phenol oxidase, various salts, amino acids, proteins, cations, lipids, ammonia, and enzymes in *S. sclerotiorum* (*Cooke, 1969*; *Jones, 1970*; *Colotelo, 1971*); these ingredients can reflect function to a certain extent. Although there has been little research on the function of exudate droplets, studies should be performed due to their physiological significance. With excess soluble carbohydrates released in a direct or converted way during effluence of exudate droplets from sclerotia, exudate droplets can maintain the internal physiological balance of sclerotia through a selective mechanism (*Cooke, 1969*). The carbohydrates in droplets may have a significant influence on long-term survival of the sclerotia (*Daly, Knoche & Wiese, 1967*; *Willetts, 1971*; *Chet & Henis, 1975*; *Willetts & Bullock, 1992*). With the exception of carbohydrates, some other constituents in exudate droplets are also reabsorbed and probably utilized by sclerotial tissues (*Colotelo, 1978*).

Sclerotial size, weight, and germination are affected by the depletion of exudate droplets during sclerotia development (*Singh et al., 2002*). Studies have shown that pathogenicity is another important property of exudate droplets, and some proteins may be involved in pathogen development and virulence (*Liang, Strelkov & Kav, 2010*). In addition, the metabolites of exudate droplets such as phenolic acids from *Rhizoctonia solani* contribute to their antifungal, phytotoxic, and antioxidant activities (*Aliferis & Jabaji, 2010*).

To gain a better understanding of exudate function, especially its role in defense against pathogens, sclerotium development and virulence, a proteome-level study was performed. To the best of our knowledge, there has been no detailed analysis about proteins in the exudates of *S. ginseng*. Moreover due to the significance of ginseng in herbal health care, it is imperative to research the exudation of *S. ginseng*. In this study, the identified proteins were classified and their functions were discussed.

## MATERIALS AND METHODS

### Fungal isolate and culture conditions

The strain (QY-6) was collected from Dasuhe Village, Qingyuan County, Liaoning Province, China on July 16, 2016. To make the separation of pathogens easier, the invaded tissues were stored at a low temperature (4 °C) for several days to induce more obvious symptoms. After isolation and purification, the pathogen was cultured on potato dextrose agar (PDA) (200 g/L potato, 20 g/L dextrose, 20 g/L agar power, sterilized) in a Petri dish, and incubated at optimum temperature (20 ± 1 °C) in the dark until it became an active mature sclerotium for the collection of exudates.

### Generation and collection of exudates

The mature sclerotia that formed on the culture medium were shaped like an inverted bowl buckled when it reached maturation, and stage droplets adhered to the surface of sclerotia in the same manner that sweat sticks to people. The agar block with mycelia was incubated at optimum temperature (20 ± 1 °C) for 5 days, after which a 5 mm diameter mycelia plug was cut along the edge of the fresh colony to start a new round of growth for the collection of exudates. The exudates were sucked from the surface of the sclerotia with a disposable blood collection tube (20 µL), and the liquid was stored in a 1.5 mL microcentrifuge tube (GEB, Torrance, CA, USA) at −20 °C for subsequent experiments. To collect a minimum amount of sclerotial exudates, approximately 600 Petri dishes with QY-6 isolate were cultured and divided into three batches, after which the droplets were collected and mixed together for further analysis.

### Protein preparation

The exudate droplets were ultrafiltered and concentrated to 300 µL, added to 1 volume of protein solution 4 volumes of cold acetone, and then mixed and kept overnight at −20 °C. We spun the mixture 10 min at 4 °C in microfuge at speed 12,000 g and then carefully discharged supernatant and retained the pellet: dried tube by inversion on tissue paper. We dried samples under dry air to eliminate any acetone residue. In order to dissolve the protein of the dry pellet, we added 50 L sample lysate (1% DTT, 2% SDS, 10% glycerine,

50 mM Tris–HCl, pH6.8) to the dry pellet at room temperature for 4 h. We spun the protein solution 15 min at room temperature in microfuge at speed 12,000 g and retained the supernatant, which contained total protein and stored the supernatant of total protein at −80 °C.

## SDS-PAGE

For one-dimensional SDS-PAGE, 30 µg protein sample was added to a 1.5 mL eppendorf tube, mixed with an equal volume of denaturing sample buffer, and placed in boiling water for 5 min and then samples were subjected to SDS-PAGE on 12% gels. Electrophoresis was performed at 80 V until the bromophenol blue had reached the separating gel, after which the voltage was increased to 120 V. After electrophoresis, the gel was stained with Coomassie Brilliant Blue G-250 for 6 h and then destained with destaining solution (25 mM $NH_4HCO_3$, 50% acetonitrile aqueous solution).

## In-gel protein digestion

Protein samples for in-gel digestion were prepared according to *Katayama, Nagasu & Oda (2001)* with some minor modifications. Protein bands were excised from the stained 1-DE gels and placed the gel pieces in a 1.5 ml eppendorf tube. We added 200 µL of the ultrapure water once time and rinsed the gel pieces twice. We then removed the ultrapure water and added 200 µL of the destaining solution (25 mM $NH_4HCO_3$, 50% acetonitrile aqueous solution) at room temperature for 30 min. We removed the destaining solution and added dehydration solution 1 (50% acetonitrile solution) for 30 min. We removed the dehydration solution 1 and added dehydration solution 2 (100% acetonitrile solution) for 30 min. We removed the dehydration solution 2 and added reduction solution 1 (25 mM $NH_4HCO_3$ and 10 mM DTT) at 57 °C for 1 h. We removed the reduction solution 1 and added reduction solution 2 (25 mM $NH_4HCO_3$ and 50 mM iodoacetamide) at room temperature for 30 min. We removed the reduction sulution 2 and added imbibition solution (25 mmol/L $NH_4HCO_3$) at room temperature for 10 min. We removed the imbibition solution and add dehydration solution 1 for 30 min. We then removed the dehydration solution 1 and added dehydration solution 2 for 30 min. The gels were rehydrated in 10 µL digest solution (0.02 µg/µL trypsin and 25 mM $NH_4HCO_3$) for 30 min and 20 µL cover solution (25 mM $NH_4HCO_3$) was added for digestion 16 h at 37 °C with occasional vortex mixing. The solutions were added to new microcentrifuge tubes, and the gels were extracted once with 50 µL extraction buffer (5% TFA and 67% acetonitrile) at 37 °C for 30 min. The extracts were thoroughly mixed with solutions and then completely dried.

## Liquid chromatography-mass spectrometry

The dried samples were redissolved in nano-HPLC buffer A (water solution containing 1% HCOOH), and separated using the nano-HPLC liquid phase EASY-nLC1000 system. The mobile phase consisted of A liquid (water solution containing 1% HCOOH) and B liquid (ACN solution containing 1% HCOOH). The samples were loaded on a trap column of 100 µm × 20 mm (RP-$C_{18}$, Thermo) with an auto-sampler and separated with an analysis column (75 µm ×150 mm, RP-$C_{18}$, Thermo) at 300 nL/min. Enzymatic hydrolysate was analyzed with a LTQ Orbitrap Velos Pro (Thermo Finnigan, Somerset, NJ,

USA), with an analysis time of 105 min and in cation detection mode. Data were acquired using an ion spray voltage of 1.8 kV and an interface heating temperature of 150 °C. The parent ion scanning range was 350–1,800 m/z. For information-dependent acquisition, the strongest 15 pieces of the MS2 scan were acquired after each full scan. Fracture mode: collision-induced dissociation, normal chemical energy was 35%, the $q$ value was 0.25, and the activation time was 30 ms. Dynamic exclusion was set for $\frac{1}{2}$ of the peak width (30 s). The mass spectrometry (MS) resolution was 60,000, while M/Z 400 and tandem MS (MS/MS) resolution were the unit mass resolving in the ion tap. The precursor was refreshed from the exclusion list. MS using profile model collection and MS/MS using the centroid method were used to collect data to reduce the file size.

### Protein identification
Based on the combined MS and MS/MS spectra, proteins were successfully identified based on the 95% confidence interval of their scores in the MASCOT V2.3 search engine (Matrix Science Ltd., London, UK), using the following search parameters: *S. sclerotiorum* database; trypsin as the digestion enzyme; two missed cleavage sites; fixed modifications of carbamidomethyl (C); partial modifications of acetyl (Protein N-term), deamidated (NQ), dioxidation (W), oxidation (M); and ±30 ppm for precursor ion tolerance and ±0.15 Da for fragment ion tolerance.

### Bioinformatic analysis
We used the NCBInr database to obtain basic information of the identified proteins from the excised gel slices. Pfam (http://pfam.sanger.ac.uk) and InterPro (http://www.ebi.ac.uk/interpro/) were analyzed to find conserved domains/regions/motifs. Functional categorization of proteins was analyzed with websites including UniProt (http://www.uniprot.org/) and the Gene Ontology database (http://www.geneontology.org/). The metabolic pathway, and which protein participated in it, was attributed to the KEGG Pathway (http://www.genome.jp/kegg/pathway.html) and EnsemblFungi (http://fungi.ensembl.org/index.html).

## RESULTS AND DISCUSSION
### Description of sclerotial exudates and protein profiles
As described in the experimental procedures, by day 6, aerial hyphae were first bestrewed on the PDA culture in a Petri dish (90 mm) at 20 °C in the dark (Fig. 1A). At the same time the aerial hyphae ceased vegetative growth and started to differentiate, which was consistent with previous reports (*Erental, Dickman & Yarden, 2008*; *Patsoukis & Georgiou, 2007*) (Figs. 1B and 2A). After 8 days of inoculation, the obvious appearance of sclerotium development was observed, with the hyphae coalesced into snowball-shaped mycelia (Figs. 1C and 2B). At the same time, with the extended incubation time, exudate droplets started to separate on the surface (Fig. 2C) and their quantity gradually increased (Fig. 2D). At about day 12, the sclerotia were characterized by surface delimitation, internal consolidation, and pigmentation, while the droplets had considerably enlarged to almost the maximum amount (Figs. 1D and 2E). By day 15, the exudate droplets had completely

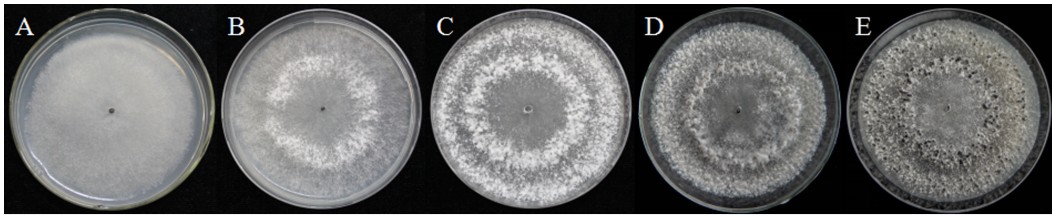

**Figure 1 Morphological characteristics of isolates QY-6 on potato dextrose agar after incubation at 20 °C in the dark.** (A) Vegetative growth of aerial hyphae. (B) Initial process of differentiation. (C) Obvious appearance of sclerotium development. (D) Collection times. (E) Maturation of sclerotia characterized by surface delimitation and pigmentation.

disappeared in several ways (Fig. 2F). It has been reported that droplets can remain on the surface of sclerotia for a period of time, but eventually disappear due to evaporation and reabsorption (*Willetts & Bullock, 1992*; *Pandey et al., 2007*). In this experiment, to collect enough exudate droplets, more than 600 Petri dishes (90 mm) with QY-6 isolate were cultured; however in other species, the collection of droplets has not been as difficult and labor-intensive (*Pandey et al., 2007*; *Aliferis & Jabaji, 2010*; *Liang, Strelkov & Kav, 2010*). This disparity was presumably caused by the distinct water metabolism ability of different pathogens (*Willetts & Bullock, 1992*). Another reason might be related to the moisture content of food source. Exudate droplets that were collected from *S. ginseng* exhibited water solubility and its color was light yellow. Sclerotia of *S. ginseng* were spherical and elongated; some were aggregated to form irregular shapes, and some were produced near the colony margin in concentric rings, firmly attached to the agar surface (Fig. 1E). The SDS-PAGE profiles showed protein gel slices in the range of 14 and 220 kDa, and the corresponding gels displayed good resolution with low background and streaking (Fig. 3).

## Protein identification by liquid chromatography-tandem mass spectrometry

After exudate droplets were fractionated and analyzed by SDS-PAGE and liquid chromatography-tandem mass spectrometry (LC-MS/MS), peptide mass fingerprints of the proteins excised from gel slices were evaluated to search the NCBInr database using Mascot software (http://www.matrixscience.com). The ion score was $-10 \log (P)$, where $P$ was the probability that the observed match was a random event. Individual ion score greater than the threshold value ($> 33$) indicated identity or extensive homology ($p < 0.05$). Because the exudate droplets of *S. ginseng* were not yet fully annotated, the MS/MS-based identification strategy was not trivial. A total of 122 non-redundant proteins were identified from the SDS-PAGE gel slices.

## Functional categories and proteome mining data

So far, the complete genome sequences of *S. ginseng* has not been studied, which leads to a certain limitation in the search for identification of exudate droplets proteome. However, the complete genome sequences of closely related species of *S. sclerotiorum* and *B. cinerea* are openly available in the database that can be used as a reliable reference to identify the

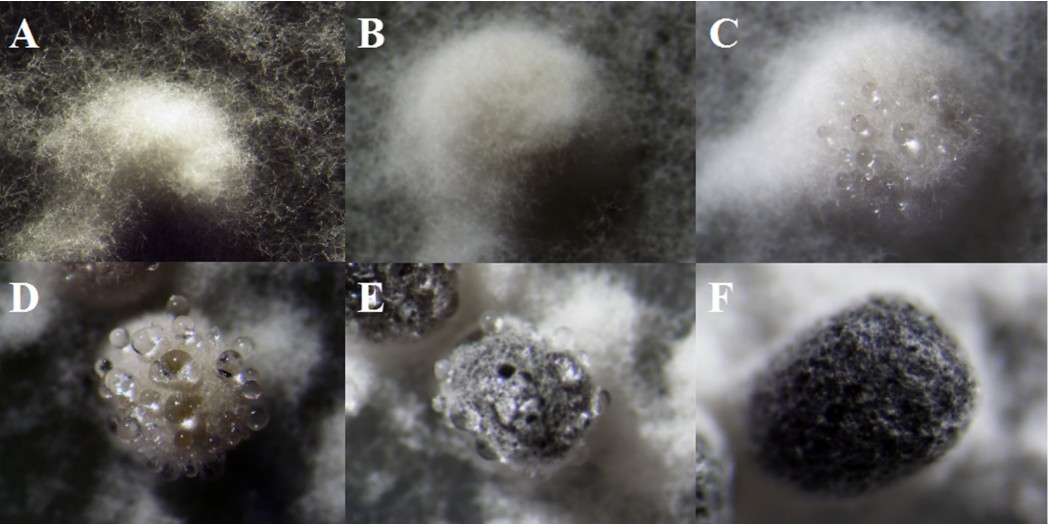

**Figure 2** **The formation of exudate droplets accompanying sclerotia development of *Sclerotinia ginseng*.** (A) Vegetative hyphae coalesced into mycelia, which represented the stage of sclerotia initial (SI). (B) Condensation of internal mycelium and enlargement of sclerotium. (C) During the earlier stage of sclerotia development (SD), exudate droplets began to emerge on the surface of the sclerotia. (D) Consolidation of sclerotium with more exudate droplets separated out of the sclerotial surface. (E) Maturation of sclerotia characterized by surface delimitation and pigmentation. (F) Sclerotia mature (SM).

exudate droplets proteome of *S. ginseng*. Based on the available search databases, a total of 122 proteins were calculated from exudation of *S. ginseng*, and only 59 proteins individual ion scores satisfied the conditions (threshold value > 33) (Table 1). The identified proteins were classified into six groups according to their characteristics based on GO terms analysis. The major functional groups were carbohydrate metabolism (39%), oxidation–reduction process (12%), transport and catabolism (5%), amino acid metabolism (3%), other functions (18%), and unknown protein (23%) (Fig. 4).

During the development of *S. ginseng*, it was not difficult to find that the development and maturation of this pathogen accompanied huge morphological changes especially in remodeling fungal cell wall. The fungal cell wall is pivotal for cell shape and function and acts as an interfacial protective barrier during host infection and environmental challenges (*Samalova et al., 2017*). Most fungal cell walls consist of a crosslinked matrix of glucans, chitins, and cell wall proteins (*Ao et al., 2016*). According to the results, proteins of the carbohydrate metabolic process (39%) of exudate droplets accounted for the largest proportion, which consistent was with a previous study (*Liang, Strelkov & Kav, 2010*). Among this group, most of them belonged to glycoside hydrolase family members, which mainly had functions of glucanase activity (gi|154703817|, gi|154701174|, gi |347832626|) and chitinase activity (gi|154691632|). Some of them also are identified with the research of Liang (*Liang, Strelkov & Kav, 2010*). *Ao et al. (2016)* demonstrate that the NGA-1 exo-chitinase and the CGL-1 β-1,3-glucanase play critical roles in remodeling the *Neurospora crassa* conidia cell walls. Chitinase is also involved in the process of cell wall formation by modifying cell wall architecture during hyphal growth in *Neurospora crassa*

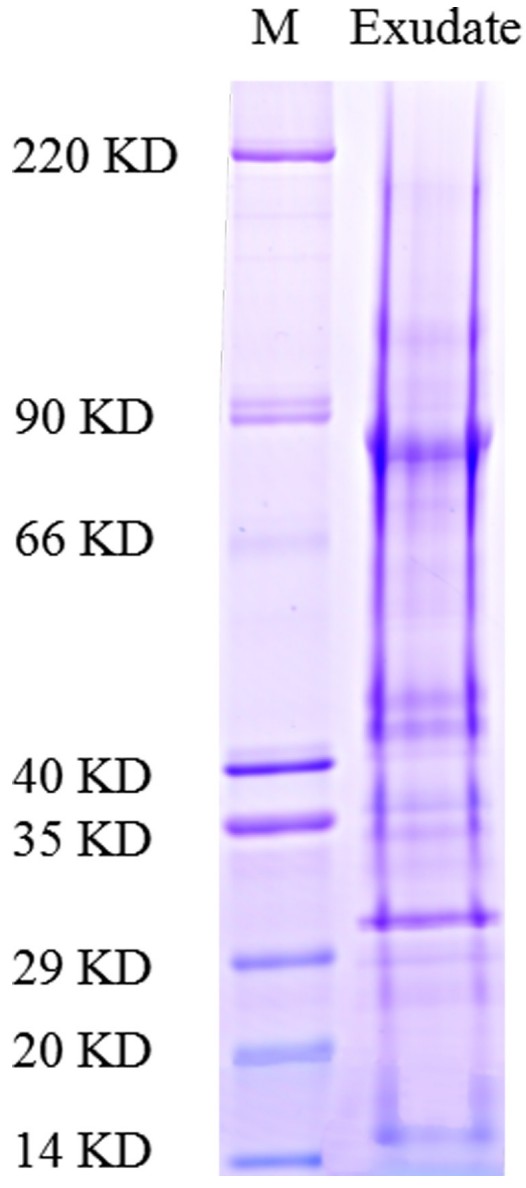

**Figure 3 Functional classification of proteins identified in sclerotial exudates from *Sclerotinia ginseng*.** Proteins were separated by SDS-PAGE and visualized by staining with colloidal Coomassie blue.

(*Tzelepis et al., 2012*). It may be suggested that some proteins in carbohydrate metabolic process group has assisted in formation of fungal cell walls. *Colotelo (1978)* also indicated that the droplets were associated with actively growing mycelia. *Samalova et al. (2017)* characterizes five putative β-1,3-glucan glucanosyltransferases play significant roles in structural modification of the cell wall of *Magnaporthe oryzae* during appressorium-mediated plant infection. Glucanosyltransferases (gi|154694741|, gi |154697112|, gi|154698335|, gi|154698875|) had also been identified in the carbohydrate metabolism group of exudate droplets and the protein (gi|154694741|) possesses the role in elongation

**Table 1   List of Proteins Identified by SDS-PAGE with LC-MS/MS in the Sclerotial Exudates.**

| GI[a] | Protein name[b] | Score[c] | Mass[d] | Matches[e] | Sequences[f] | emPAI[g] |
|---|---|---|---|---|---|---|
| | **Carbohydrate metabolic process** | | | | | |
| gi\|347830055\| | glycoside hydrolase family 92 protein | 513 | 86,256 | 16 (11) | 7 (4) | 0.16 |
| gi\|154705171\| | hypothetical protein SS1G_07393 | 260 | 83,872 | 12 (8) | 7 (3) | 0.12 |
| gi\|154703817\| | glucan 1,3-beta-glucosidase SS1G_06037 | 231 | 46,611 | 20 (10) | 9 (6) | 0.61 |
| gi\|154695005\| | glucoamylase SS1G_10617 | 225 | 72,488 | 11 (8) | 4 (2) | 0.09 |
| gi\|1095456302\| | glucoamylase sscle_10g080270 | 212 | 67,612 | 8 (4) | 5 (2) | 0.1 |
| gi\|154694741\| | 1,3-beta-glucanosyltransferase SS1G_10353 | 198 | 47,929 | 12 (7) | 7 (4) | 0.3 |
| gi\|154694427\| | beta-hexosaminidase SS1G_10038 | 161 | 64,156 | 13 (6) | 6 (4) | 0.22 |
| gi\|154702253\| | alpha-1,2-Mannosidase SS1G_04468 | 123 | 58,064 | 6 (5) | 4 (4) | 0.25 |
| gi\|154701174\| | hypothetical protein SS1G_03387 | 117 | 63,029 | 3 (3) | 2 (2) | 0.11 |
| gi\|347832626\| | glycoside hydrolase family 71 protein | 102 | 81,772 | 2 (1) | 2 (1) | 0.04 |
| gi\|154702326\| | hypothetical protein SS1G_04541 | 84 | 78,092 | 11 (5) | 7 (4) | 0.18 |
| gi\|154698322\| | hypothetical protein SS1G_12917 | 83 | 43,069 | 5 (2) | 2 (2) | 0.16 |
| gi\|154697112\| | hypothetical protein SS1G_01776 | 71 | 49,311 | 3 (3) | 2 (2) | 0.14 |
| gi\|154698335\| | glycoside hydrolase family 17 protein SS1G_12930 | 63 | 31,996 | 3 (2) | 2 (2) | 0.22 |
| gi\|347836311\| | glycoside hydrolase family 3 protein | 56 | 94,046 | 7 (3) | 5 (3) | 0.11 |
| gi\|154691568\| | transaldolase SS1G_00709 | 53 | 31,398 | 1 (1) | 1 (1) | 0.11 |
| gi\|154698875\| | hypothetical protein SS1G_13472 | 48 | 60,572 | 2 (1) | 2 (1) | 0.05 |
| gi\|507414638\| | carbohydrate-Binding Module family 20 protein | 46 | 43,095 | 3 (1) | 2 (1) | 0.08 |
| gi\|154693234\| | hypothetical protein SS1G_08837 | 40 | 44,896 | 1 (1) | 1 (1) | 0.07 |
| gi\|154693514\| | hypothetical protein SS1G_09118 | 38 | 33,973 | 1 (1) | 1 (1) | 0.1 |
| gi\|347830059\| | glycoside hydrolase family 28 protein | 38 | 39,845 | 2 (2) | 1 (1) | 0.08 |
| gi\|154691632\| | hypothetical protein SS1G_00773 | 36 | 19,0114 | 6 (1) | 4 (1) | 0.02 |
| gi\|347836319\| | glycoside hydrolase family 3 protein | 35 | 95,468 | 3 (1) | 2 (1) | 0.03 |
| | **Unknown protein** | | | | | |
| gi\|1095449307\| | hypothetical protein sscle_01g010410 | 1,206 | 87,452 | 74 (51) | 8 (7) | 0.55 |
| gi\|238477235\| | developmental-specific protein Ssp1 | 915 | 35,067 | 86 (45) | 17 (13) | 5.09 |
| gi\|1095455875\| | hypothetical protein sscle_10g076000 | 181 | 11,2854 | 10 (8) | 6 (4) | 0.15 |
| gi\|154696190\| | predicted protein SS1G_12133 | 170 | 37,138 | 16 (7) | 7 (5) | 0.53 |
| gi\|154705040\| | hypothetical protein SS1G_07262 | 163 | 23,894 | 6 (4) | 2 (2) | 0.3 |
| gi\|154696764\| | hypothetical protein SS1G_01428 | 105 | 45,622 | 6 (6) | 2 (2) | 0.15 |
| gi\|154704206\| | hypothetical protein SS1G_06426 | 95 | 95,078 | 6 (4) | 4 (2) | 0.07 |
| gi\|1095455126\| | hypothetical protein sscle_08g068530 | 91 | 62,168 | 9 (3) | 6 (2) | 0.11 |
| gi\|347831150\| | hypothetical protein BofuT4_P115530.1 | 67 | 41,337 | 1 (1) | 1 (1) | 0.08 |
| gi\|154700524\| | hypothetical protein SS1G_14133 | 60 | 32,906 | 6 (3) | 3 (1) | 0.1 |
| gi\|347827005\| | hypothetical protein BofuT4_P073140.1 | 47 | 66,587 | 2 (2) | 2 (2) | 0.1 |
| gi\|347838722\| | hypothetical protein BofuT4_P123130.1 | 40 | 21,674 | 2 (2) | 1 (1) | 0.16 |
| gi\|154691708\| | hypothetical protein SS1G_00849 | 38 | 16,672 | 2 (2) | 1 (1) | 0.2 |
| gi\|154693400\| | predicted protein SS1G_09003 | 33 | 7259 | 3 (1) | 1 (1) | 0.49 |

**Table 1** (*continued*)

| GI[a] | Protein name[b] | Score[c] | Mass[d] | Matches[e] | Sequences[f] | emPAI[g] |
|---|---|---|---|---|---|---|
| | **Oxidation–reduction process** | | | | | |
| gi\|154697664\| | hypothetical protein SS1G_11927 | 448 | 70,689 | 14 (11) | 4 (4) | 0.37 |
| gi\|1095450409\| | hypothetical protein sscle_02g021420 | 160 | 64,659 | 8 (4) | 6 (3) | 0.16 |
| gi\|154696912\| | hypothetical protein SS1G_01576 | 125 | 66,177 | 8 (4) | 3 (2) | 0.16 |
| gi\|347840672\| | similar to extracellular dihydrogeodin oxidase/laccase | 79 | 65,066 | 11 (7) | 4 (1) | 0.16 |
| gi\|154704145\| | hypothetical protein SS1G_06365 | 72 | 63,251 | 7 (2) | 3 (2) | 0.11 |
| gi\|347841076\| | similar to cellobiose dehydrogenase | 61 | 61,015 | 3 (2) | 1 (1) | 0.05 |
| gi\|154702896\| | laccase SS1G_05112 | 44 | 66,133 | 9 (3) | 4 (3) | 0.16 |
| | **Transport and catabolism** | | | | | |
| gi\|154693130\| | actin SS1G_08733 | 177 | 41,841 | 15 (8) | 9 (5) | 0.46 |
| gi\|1095457170\| | hypothetical protein sscle_12g088930 | 41 | 24,866 | 4 (1) | 3 (1) | 0.13 |
| gi\|154703678\| | hypothetical protein SS1G_05898 | 36 | 89,774 | 4 (1) | 1 (1) | 0.04 |
| | **Amino Acid Metabolism** | | | | | |
| gi\|154693286\| | hypothetical protein SS1G_08889 | 525 | 81,000 | 37 (23) | 11 (9) | 0.74 |
| gi\|154691589\| | hypothetical protein SS1G_00730 | 78 | 66,638 | 10 (6) | 4 (2) | 0.1 |
| | **Protein binding** | | | | | |
| gi\|1095454060\| | hypothetical protein sscle_07g057880 | 42 | 42,363 | 2 (1) | 1 (1) | 0.08 |
| gi\|154703307\| | hypothetical protein SS1G_05524 | 39 | 14,1121 | 8 (3) | 5 (1) | 0.02 |
| | **N-acetyltransferase activity** | | | | | |
| gi\|347838742\| | hypothetical protein BofuT4_P123330.1 | 52 | 27,295 | 7 (5) | 2 (1) | 0.12 |
| | **Phospholipid biosynthetic process** | | | | | |
| gi\|347827354\| | similar to phosphatidylserine decarboxylase | 43 | 46,590 | 8 (1) | 6 (1) | 0.07 |
| | **Chitin binding** | | | | | |
| gi\|154699986\| | hypothetical protein SS1G_02582 | 463 | 82,732 | 17 (14) | 9 (9) | 0.42 |
| | **ATP binding** | | | | | |
| gi\|347830591\| | similar to calcium-transporting P-type ATPase | 35 | 11,9114 | 8 (1) | 7 (1) | 0.03 |
| | **Ubiquitin-protein transferase activity** | | | | | |
| gi\|347837081\| | similar to ubiquitin-protein ligase | 37 | 13,4855 | 3 (1) | 3 (1) | 0.02 |
| | **Nucleic acid binding** | | | | | |
| gi\|154693707\| | hypothetical protein SS1G_09311 | 37 | 78,216 | 2 (1) | 2 (1) | 0.04 |
| | **Mycotoxin biosynthetic process** | | | | | |
| gi\|154696502\| | predicted protein SS1G_01165 | 54 | 17,311 | 10 (7) | 2 (1) | 0.2 |
| | **Energy metabolism** | | | | | |
| gi\|154703202\| | hypothetical protein SS1G_05418 | 34 | 10,468 | 2 (1) | 1 (1) | 0.33 |

**Notes.**

[a]Accession number in NCBI database.

[b]Protein name based on Broad database searching by Mascot results.

[c]Protein scores were derived from ions scores as a non-probabilistic basis for ranking protein families (threshold score > 33).

[d]Nominal mass (Mr).

[e]Number of matched peptides (number of matched peptides which significance threshold $p < 0.05$).

[f]Number of non-redundant matched peptides (number of non-redundant matched peptides which significance threshold $p < 0.05$).

[g]Protein abundance value.

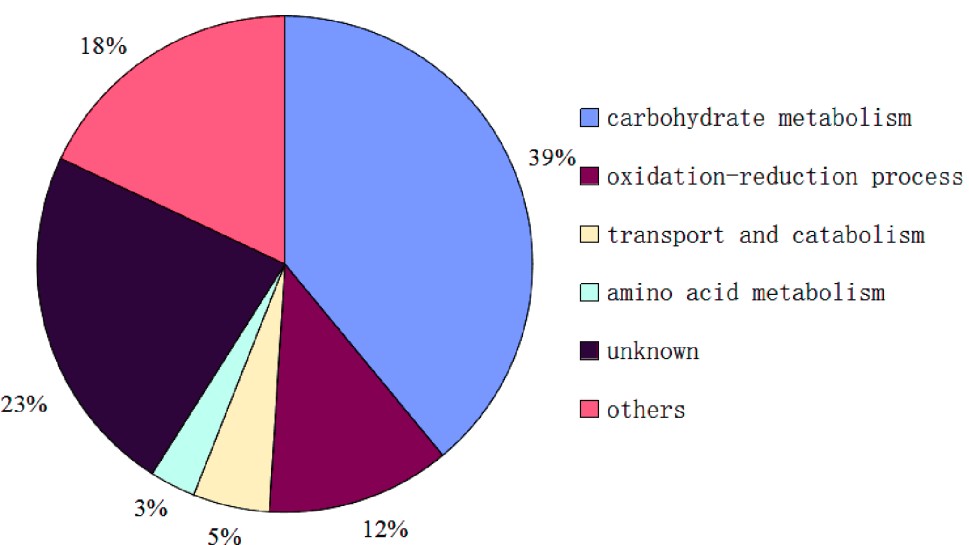

**Figure 4** Functional classification of proteins identified in sclerotial exudates from *Sclerotinia ginseng*.

of 1,3-beta-glucan chains (*Mouyna et al., 2000*) which further certifies the role in modifying cell wall structure. Some other proteins including $\alpha$-mannosidase (gi |347830055|, gi|154702253|) (*Rajesh et al., 2014*) and rhamnosidase (gi|154702326|) identified in this group were also reported to participate in modifying cell wall architecture (*Ichinose, Fujimoto & Kaneko, 2013*). As aforementioned, most proteins in this group participated in formation of the fungal cell wall (*Martens-Uzunova et al., 2006*; *Tzelepis et al., 2012*; *Ao et al., 2016*; *Samalova et al., 2017*). The amount of carbohydrate compounds might be related to the regulation of coenzymes, which could continue oxidizing the sugar material to provide energy and carbon skeleton for the development of sclerotia.

Additionally, two proteins were identified to occupy pectinase activity (gi|154705171|, gi|347830059|) and three proteins played a role in cellulase activity (gi |347836311|, gi|154693234|, gi|347836319|). Cellulose, hemicellulose, and pectin are three main components of the plant cell wall that compose an important barrier against pathogen attack. To succeed in infecting plants, fungi possess a diverse array of secreted enzymes (e.g., pectinase, cellulase) to depolymerize the main structural polysaccharide of cell wall (*Kubicek, Starr & Glass, 2014*). It is suggested that these five proteins may be in existence as a virulence factor factor of this fungus. Cellulolytic and polygalacturonase activities for the exudate and sclerotial extracts were also demonstrated in a previous study (*Colotelo, Sumner & Voegelin, 1971*). The observation can be further speculated that exudate droplets of the fungi possess pathogenicity which is involved in the maceration and soft-rotting of plant tissue.

Accounting for the second largest protein component was the unknown protein group (23%); thus, these proteins should not be overlooked as they were present in significant amounts. In this category, there was a development-specific protein (Ssp1) (gi |238477235|), which also appeared in the exudate droplets of *S. sclerotiorum* (*Liang, Strelkov & Kav, 2010*).

Although its function is unknown, it is the most abundant soluble protein in sclerotia and apothecia of *S. sclerotiorum* (*Li & Rollins, 2010*). The *ssp1* transcript accumulates exclusively within developing sclerotium tissue and not in any other examined stage of growth or development (*Li & Rollins, 2009*). It can be speculated that Ssp1 maybe a critical protein during the later stage of sclerotial development in Sclerotiniaceae family. Outside the Sclerotiniaceae *ssp1* homologs are found only from the sclerotium-forming *Aspergillus* species *A. flavus* and *A. oryzae* (*Li & Rollins, 2009*), which further illustrates the characteristic of it. Therefore, additional studies concerning the regulation, function, and mechanism of this protein would lead to a better understanding of sclerotium development. Currently, there has been increasing research about unknown functions, as these studies can provide a basis for understanding the regulation of biological processes.

Another major category that was identified was oxidation–reduction process (12%), which is an important part of the exudate droplet proteome. Four proteins were identified or similar to laccase (gi|154697664|, gi|347840672 |, gi|154704145|, gi|154702896|) and has been shown to be mainly involved in lignin biodegradation, fungal virulence, morphogenesis and melanin synthesis (*Coman et al., 2013*). In the exudate droplets of *S. sclerotiorum* only laccase (gi|154702896|) had been identified. Tyrosinase (gi|154696912|) was also found in this group, which has a critical role in formation of pigments such as melanin and other polyphenolic compounds (*Stevens et al., 1998*; *Mahendra Kumar et al., 2011*). Melanin, which plays an important role in the process of sclerotium formation, has been extensively studied, and allows the dormancy of fungal propagules by protecting them against a variety of unfavorable conditions (*Butler, Gardiner & Day, 2005*). It is also a critical factor affecting the pathogenicity of pathogens (*Butler, Gardiner & Day, 2005*; *Abo Ellil, 1999*).

A small proportion but equally important proteome can not be ignored, which also possesses significant functions. In mycotoxin biosynthetic process, oxidase ustYa is involved in the the production of ustiloxins, toxic cyclic peptides, in filamentous fungi, which wasn't identified in the exudate droplets of *S. sclerotiorum*. Ustiloxins are found recently to be the first example of cyclic peptidyl secondary metabolites that are ribosomally synthesized in filamentous fungi (*Nagano et al., 2016*), which can be suggested that exudate droplets of *S. ginseng* may participate in the biosynthesis of toxin or involve in pathogenicity and virulence.

Although most proteins of exudate droplets of *S. sclerotiorum* (*Liang, Strelkov & Kav, 2010*) and *S. ginseng* were calculated from the available search database of *S. sclerotiorum*, there were only 15 proteins presented in the two identified proteomes and most of them belonged to carbohydrate metabolic process. It was suggested that although the two pathogens all belonged to Sclerotiniaceae, but the difference was still a considerable quantity. Therefore, it is of great significance to study the exudate droplets of *S. ginseng*. With the further deepening of the research on Sclerotiniaceae, the function of some unknown proteins has been gradually revealed, which can be reflected on the proportion of unknown protein group in the exudate droplets. Compared to the unknown proteins (32%) of exudate droplets in *S. sclerotiorum* (*Liang, Strelkov & Kav, 2010*), the proportion of unknown proteins of exudate droplets in *S. ginseng* had decreased to 23%. According to our

results, more abundant categories were classified in other proteome group including protein binding, N-acetyltransferase activity, phospholipid biosynthetic process, chitin binding, ATP binding, ubiquitin-protein transferase activity, nucleic acid binding, mycotoxin biosynthetic process and energy metabolism, which could reflect functional diversity of exudate droplets indirectly.

In conclusion, future studies should be performed on the metabolic pathway and functions of the exudate droplets using modern techniques. In addition, because the proportion of unknown protein components was still relatively high in this study, they should also be the focus of future studies.

### Funding
This work was supported by the Special Fund for Agroscientific Research in the Public Interest of China (No. 201303111). The funders had no role in study design, data collection and analysis, decision to publish, or preparation of the manuscript.

### Grant Disclosures
The following grant information was disclosed by the authors:
Agroscientific Research in the Public Interest of China: 201303111.

### Competing Interests
The authors declare there are no competing interests.

### Author Contributions
- Dan Wang conceived and designed the experiments, performed the experiments, analyzed the data, contributed reagents/materials/analysis tools, wrote the paper, prepared figures and/or tables, reviewed drafts of the paper.
- Jun Fan Fu conceived and designed the experiments, contributed reagents/materials/-analysis tools, reviewed drafts of the paper.
- Ru Jun Zhou contributed reagents/materials/analysis tools, reviewed drafts of the paper.
- Zi Bo Li and Yu Jiao Xie contributed reagents/materials/analysis tools.

### Data Availability
The raw data has been provided as a Supplemental File.

### Supplemental Information
Supplemental information for this article can be found online at http://dx.doi.org/10.7717/peerj.3979#supplemental-information.

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
