# Peer review of "Proteomics research and related functional classification of liquid sclerotial exudates of Sclerotinia ginseng"

_PeerJ, doi:10.7717/peerj.3979_

## Round 0.1 · original submission · Major Revisions

· Academic Editor

Major Revisions

Reviewer 1 in particular has taken great care in critiquing your work and I find this review very comprehensive. The comments on materials and methods were spot on. You may also want to seek some English usage program to better convey your work.

Reviewer 1 ·

Basic reporting

I found that the manuscript had a large number of sentences that need to be rewritten in clear and unambiguous English, and would recommend that the authors get some help in writing the manuscript. Some of the sentences that I found to be unclear can be found on lines 38, 121,123, 150, 174, 190, 276, and 290.

I found that the materials and methods section was very difficulty to follow. In multiple instances important information was lacking. For example, it's unclear whether multiple gel slices were used for the proteomic analysis or if the entire protein mixture was included in a single analysis. Items like the composition of Buffer B (line 150) or what's in the solutions 1, 2 and extraction buffer (lines 139 - 145) need to be added. To summarize, the entire materials and methods section needs to be rewritten.

The authors did not include all of the references from the text in the References. For example they cite an article from Georgios et al, 2012 (lines 254 and 315) on Neurospora chitinases, and don't include the reference in their references cited. In line 315 the reference is given as evidence that the chitinases are used during sclerotia formation. While I don't doubt that chitinases are expressed during sclerotia formation, the reference is for Neurospora crassa, which does not make sclerotia - the reference doesn't apply to the statement being made and should be deleted from line 315.

I would also strongly request that the authors include a figure of the gel electrophoresis. This is a vital piece of data that needs to be included.

Another very important element that is lacking is a much more detailed analysis and discussion of how their results for S. ginseng exudates compares to analysis of Liang et al on the proteins in S. sclerotiorum exudates. The authors practically ignore the excellent proteomic analysis for S. sclerotiorum exudates even though they are using the S. sclerotiorum genome as their basis for identifying their proteins. A close examination of the Liang et al paper will be instructive to the authors on how they need to present their findings. My examination of their findings would indicate they haven't found anything new or novel.

I also think that the authors need to re-examine their discussion section and the references they are using to make sure they more accurately present the research literature and how the findings add to that literature. For example, the work of Colotelo and it's relationship to their work would be a good topic to discuss.

Experimental design

It is difficult to assess the experimental design for the proteomic analysis because the materials and methods section lacks key information about how the experiments were conducted. The basic approach is sound and the type of information they obtained seems to be what one would expect. The research question is well defined but a more rigorous comparison with what is known about exudates in other Sclerotinia species would be helpful. One major flaw in the analysis is that the S. ginseng genome is not available and so the S. sclerotiourm data-base was used. Although I understand that they couldn't use the data-base for the organism being studied, the problems with using another data-base are not discussed or addressed in any way.

Validity of the findings

The data does seem to be robust and statistically sound.

Additional comments

Although I think a proteomic analysis of Sclerotina exudates is a good research project, I can not recommend that the article be published. I would recommend a total rewriting of the article with the inclusion of key information that is currently lacking (like the gel picture, and a comparison to other exudates). I think the materials and methods section needs to be completely rewritten to include a clearer narrative of how the experiments were performed. The current discussion section is lacking key analyses. As mentioned above, the Liang et al article covers the same type of analysis and would be a good model for the authors to follow.

Reviewer 2 ·

Basic reporting

The article is sufficiently concise and well-written, and provides necessary background an experimental details.

The raw data has not been shared. At a minimum, the peptide sequences should be made available for the matched peptides.

Experimental design

Experimental design is acceptable. Although there are no independent replicates, the objective is merely to identify proteins present, not quantify them. Exudates from >600 plates were collected, so this provides some pseudoreplication that is sufficient for this purpose.

Validity of the findings

Findings are valid and sound, except as noted in the General comments for the author (below).

Additional comments

If the journal format allows it, the Results & Discussion sections should be combined.

In any case, most of the Discussion is not relevant, and does not draw on or reference the Results. For example, lines 300-311 (about the droplet morphology) are in fact new results that were not presented in the results section. Regardless, the information lines 300-311 is not particularly useful or relevant, or well-supported and thus should be removed. Lines 287-293 provide very general information about proteins and could also be deleted.

A list of peptide sequence for the identified peptides should be provided, if not the raw spectra.

---

## Round 0.2 · Minor Revisions

· Academic Editor

Minor Revisions

If you can please check the Bradford assay as indicated by reviewer 1,

Reviewer 1 ·

Basic reporting

The revised manuscript has been greatly improved. I find the English and the referencing of the research literature to be acceptable.

Experimental design

The revised Material and Methods section has been rewritten and is much improved. The only comment I have is that the authors used a Bradford assay to determine the protein concentration in their concentrated exudate sample (line 118), which contained 2% SDS. Generally the Bradford is not used for protein determination with SDS containing samples since SDS affects the assay. However, they might have used the Bradford assay and included SDS in their standard curve. I would simply ask the authors to double check whether they used the Bradford or some other assay for protein determination and if the Bradford assay was done and the standard curve was generated in an SDS solution to state that this was the case.

Validity of the findings

The data presented clearly identifies a number of proteins in the S. ginseng exudate. Since a similar study with S. sclerotiorum exudates has been done, I do not see their findings as being especially novel, but their report does add to the information about fungal exudates. Their conclusions are well stated and supported.

Additional comments

The manuscript has been substantially improved by the rewriting process.

Reviewer 2 ·

Basic reporting

Revised manuscript is now acceptable.

Experimental design

Revised manuscript is now acceptable.

Validity of the findings

Revised manuscript is now acceptable.

---

## Round 0.3 · accepted · Accept

· Academic Editor

Accept

Thank you for your revision